# Learnable Negative Proposals Using Dual-Signed Cross-Entropy Loss for Weakly Supervised Video Moment Localization

## ABSTRACT

Most existing methods for weakly supervised video moment localization use rule-based negative proposals. However, the rule-based ones have a limitation in capturing various confusing locations throughout the entire video. To alleviate the limitation, we propose learning-based negative proposals which are trained using a dual-signed cross-entropy loss. The dual-signed cross-entropy loss is controlled by a weight that changes gradually from a minus value to a plus one. The minus value makes the negative proposals be trained to capture query-irrelevant temporal boundaries (easy negative) in the earlier training stages, whereas the plus one makes them capture somewhat query-relevant temporal boundaries (hard negative) in the later training stages. To evaluate the quality of negative proposals, we introduce a new evaluation metric to measure how well a negative proposal captures a poorly-generated positive proposal. We verify that our negative proposals can be applied with negligible additional parameters and inference costs, achieving state-of-the-art performance on three public datasets.

## CCS CONCEPTS

• **Computing methodologies → Visual content-based indexing and retrieval**.

## KEYWORDS

video moment localization, learning-based negative proposal, dual-signed cross-entropy loss, evaluation metric

## 1 INTRODUCTION

Given a natural language sentence query and a video, video moment localization aims to locate a precise temporal boundary of a video segment corresponding to the sentence query. As it enables automatic extraction of relevant video segments according to given sentences, video moment localization has attracted much attention in recent years and has a wide range of applications such as video retrieval [12], visual question answering [1, 39], and video summarization [30]. Fully supervised methods [13, 48, 49] have shown impressive results but need manually annotated temporal boundaries for every pair of a video and a sentence for training, which is time-consuming and labor-intensive.

On the other hand, weakly supervised methods [36, 38, 51] only need pairs of a video and a sentence for training. Therefore, it is

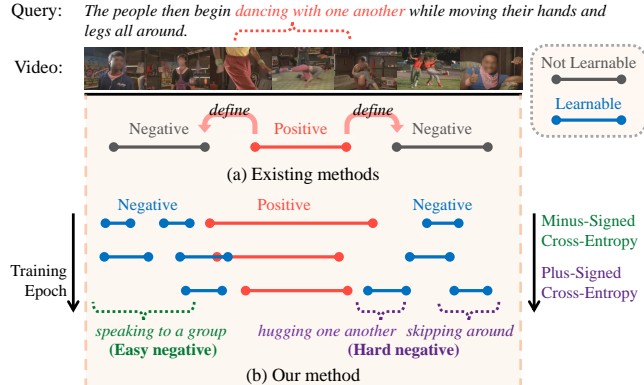

**Figure 1: Weakly supervised video moment localization. (a) Existing methods generate rule-based negative proposals depending on a positive proposal. (b) The proposed method generates learning-based negative proposals trained by a dual-signed cross-entropy loss to capture various confusing locations.**

much easier to collect a large amount of data for training, because the video-sentence pairs can be obtained from metadata on the Internet or through automatic speech recognition (ASR) [31]. Most weakly supervised methods adopt a two-stage approach that generates a positive proposal representing a specific temporal location and then employs this proposal to find a temporal boundary. To effectively generate a positive proposal, Multiple Instance Learning (MIL)-based methods make negative proposals and use contrastive learning to distinguish the positive proposal from the negative proposals. Some MIL-based methods [14, 45, 50, 57] make negative proposals from other videos that do not match a given sentence query (*i.e.*, unmatched videos). However, these negative proposals in unmatched videos are not hard enough because confusing locations usually exist in a video that matches the given sentence query (*i.e.*, a matched video).

Considering the tendency that negative proposals in the matched video are more confusing than those in the unmatched video, previous methods [15, 51, 58, 59] rely on rule-based negative proposals inside the matched video. Specifically, in [51, 58], a negative proposal is defined as a temporal location that is not captured by a positive proposal. In [15, 59], two negative proposals are set as two Gaussians whose location is predefined outside both sides of a positive proposal. These rule-based negative proposals are only determined from a positive proposal with heuristic rules. Therefore, these negative proposals have limitations in capturing various confusing locations.

To alleviate the limitations, we propose learning-based negative proposals for weakly supervised video moment localization,

which have been overlooked in the previous methods. To this end, we leverage a novel dual-signed cross-entropy loss to learn negative proposals that are gradually changed from easy to hard ones. Specifically, we generate multiple negative proposals whose center and width are learnable and select negative proposals with different weights. Then, we predict new sentence queries from the selected negative proposals and compare the predicted queries to the original query via the cross-entropy losses. We then multiply the losses by a weight value for the dual-signed cross-entropy loss (named 'cross-entropy weight' for simplicity), which is scheduled to increment from a minus value to a plus value as the training epoch progresses. According to the scheduled cross-entropy weight, our negative proposals are trained by two processes: 1) in a minus cross-entropy weight, the deconstruction process works to maximize the cross-entropy losses to learn easy negative proposals capturing a query-irrelevant temporal boundary; 2) in a plus cross-entropy weight, the reconstruction process works to minimize the cross-entropy losses to learn hard negative proposals capturing a somewhat query-relevant temporal boundary. During both processes, we leverage multiple contrastive losses to discriminate a positive proposal from multiple negative proposals. To validate the quality of negative proposals, we propose a new evaluation metric, Intersection of Negative duration, which measures how well a negative proposal captures a poorly-generated positive proposal. Our experiments are conducted on Charades-STA [13], ActivityNet Captions [23], and TV show Retrieval [25]. In summary, our contributions are as follows.

- In contrast to previous rule-based negative proposals, we propose negative proposals that are 1) learnable, 2) softly selected, and 3) gradually changed from easy to hard ones, which are trained by a dual-signed cross-entropy loss, to capture various confusing locations in a video.
- We introduce a new evaluation metric that measures how well a negative proposal captures a poorly-generated positive proposal and verify that our negative proposals have better quality than the previous negative proposals.
- We demonstrate our negative proposals significantly boost the performance of the existing methods with negligible additional parameters and inference costs, achieving state-of-the-art performance on three public datasets.

## 2 RELATED WORK

**Weakly supervised video moment localization.** Most of the weakly supervised video moment localization methods can be grouped into two categories: reconstruction-based methods and multiple instance learning-based methods. Reconstruction-based methods focus on generating positive proposals that reconstruct a sentence query. Lin *et al.* [28] introduce a sentence query reconstruction approach and uses sliding windows as positive proposals. Further, in [6, 15, 20, 22, 29, 51, 58, 59], Gaussian functions are utilized to generate learnable proposals. To refine proposals, some methods [3, 4, 6] distill other knowledge into proposals. However, the previous methods only focus on generating positive proposals. For better quality of positive proposals, negative proposals also play an important role to be used for contrastive learning with positive proposals. Therefore, we focus on generating negative proposals

and propose learning-based negative proposals using a dual-signed cross-entropy loss.

**Multiple Instance Learning (MIL).** Multiple instance learning (MIL) has been widely used in many weakly-supervised video-level computer vision problems [2, 26, 37]. MIL-based weakly supervised moment localization methods [8, 14, 15, 45, 50, 51, 56–59] make negative proposals that do not correspond to the sentence query to distinguish a positive proposal from the negative proposals. Some methods [14, 45, 50, 57] make negative proposals from other videos that do not match the query. Moreover, Chen *et al.* [8] create pseudo labels from unmatched videos. However, these negative proposals in unmatched videos are not hard enough because confusing video locations are usually inside the same video that matches the query. To consider negative proposals in a matched video, which are more confusing than negative proposals in unmatched videos, some methods [15, 51, 58, 59] make rule-based negative proposals inside the matched video. In [51, 58], a negative proposal is a temporal location that is not captured by a positive proposal. In [15, 59], two negative Gaussian proposals whose locations are predefined outside both sides of a positive proposal are used. These rule-based negative proposals are defined by a positive proposal and thus have limitations in capturing various confusing locations. To alleviate the limitations, we propose learning-based negative proposals that are trained by a dual-signed cross-entropy loss.

**Curriculum learning.** Curriculum learning is a training strategy that trains a model from easy data to hard data gradually. Conventional curriculum learning [5, 17, 18] is a training strategy that uses data with low training loss at early training. Curriculum learning has been used in many computer vision problems such as object detection [27] and video moment localization [24, 47, 59]. For fully supervised video moment localization, Lan *et al.* [24] create a negative proposal through three video data augmentations and apply each augmentation at the pre-defined time step. However, this negative proposal is based on simple rules and is not a truly-gradual curriculum design. For weakly supervised moment localization, in [59], the size and location of two negative proposals are defined by a positive proposal and controlled slightly by the current training epoch. However, in this method, the negative proposals are rule-based and depend on the positive proposals, which can not capture diverse confusing locations. Unlike previous curriculum designs, we control cross-entropy losses of negative proposals by a weight changing gradually from a minus value to a plus one during training. Through our novel loss, negative proposals can be trained to be query-irrelevant at a minus value (easy negative) and then somewhat query-relevant at a plus value (hard negative). Therefore, our learning-based negative proposals can capture diverse confusing locations, which can be exploited for effective contrastive learning.

## 3 PROPOSED METHOD

**Problem setting.** In weakly-supervised video moment localization, our goal is to locate a temporal boundary of a video segment corresponding to a sentence query without any ground-truth temporal boundary at training. Reconstruction-based methods [6, 15, 28, 36, 51, 58, 59] generate a positive proposal and train the positive proposal to reconstruct the original sentence query from a masked sentence query. These methods assume that a positive proposal

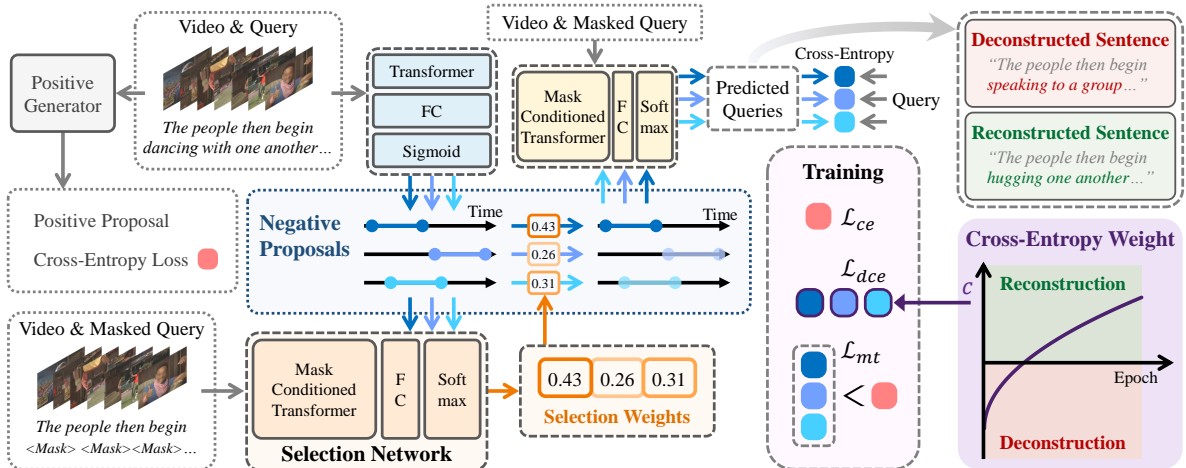

**Figure 2: Overall architecture of our method to generate negative proposals. One positive proposal and multiple negative proposals are generated from the features of a video and a query. Through a selection network, we select useful negative proposals for query prediction. Then, a new sentence query is predicted from a video, a randomly masked query, and each negative proposal. For a dual-signed cross-entropy loss $\mathcal{L}_{dce}$, we compute a cross-entropy loss between the predicted query and the original query and then multiply the cross-entropy loss by a cross-entropy weight $c$ depending on the training epoch. As the cross-entropy weight changes, the negative proposals are trained to be gradually changed from easy to hard ones. Finally, we utilize multi-triplet loss $\mathcal{L}_{mt}$ for contrastive learning.**

reconstructing the query well can be a temporal boundary corresponding to the sentence query. We follow this assumption and use the existing reconstruction-based network to generate a positive proposal. However, unlike the previous methods, we focus on generating learning-based negative proposals. We use negative proposals to predict multiple queries from a masked query and then exploit the predicted queries to calculate our dual-signed cross-entropy loss.

**Overview.** The overall architecture of our method to generate negative proposals is depicted in Fig. 2. For a positive proposal, we use an existing network [20, 58, 59] as a positive generator. For negative proposals, we utilize features extracted from a video and a sentence query as input to a transformer [41] to estimate centers and widths of multiple proposals. Then, we select useful negative proposals for query prediction through a selection network. For query prediction, we utilize a video feature, a randomly masked sentence query feature, and selected negative proposals to predict new sentence queries. The predicted query from each negative proposal is compared to the original query through a cross-entropy loss. Hence, multiple cross-entropy losses for multiple negative proposals are calculated.

To create negative proposals that are gradually changed from easy to hard ones, we leverage a novel dual-signed cross-entropy loss. First, we multiply the cross-entropy losses by a weight value $c$ scheduled to increment from a minus value to a plus value as the training epoch progresses. According to $c$, our negative proposals are trained through two processes: the deconstruction process ($c < 0$) and the reconstruction process ($c > 0$). (1) The deconstruction process maximizes the cross-entropy losses, which learn easy negative proposals to capture a query-irrelevant temporal boundary. (2) The reconstruction process minimizes the cross-entropy

losses, which learns hard negative proposals to capture a somewhat query-relevant temporal boundary. During both processes, we utilize multiple contrastive losses to discriminate the positive proposal from multiple negative proposals.

### 3.1 Feature Extraction

We extract a video feature $\mathbf{V} \in \mathbb{R}^{T \times C}$ from a video via the pretrained 3D Convolutional Neural Networks [7, 40], where $T$ is the number of sampled segments and $C$ is the feature dimension. We extract a query feature $\mathbf{Q} \in \mathbb{R}^{L \times C}$ from a sentence query via the pre-trained GloVe [33], where $L$ is the sentence length.

### 3.2 Negative Proposal Generation

**Learnable proposal generation.** Inspired by Gaussian-shaped positive proposals [6, 15, 51, 58, 59], we adopt a Gaussian shape for learning-based negative proposals, where our novel dual-signed cross-entropy loss is applied. First, we use transformer [41] to obtain multi-modal features from a video feature $\mathbf{V}$ and a query feature $\mathbf{Q}$. We append a learnable token to $\mathbf{V}$, which is a [CLASS] token in [10]. Given $\mathbf{Q}$ and $\mathbf{V}$, transformer outputs $\{\mathbf{o}_t\}_{t=1}^{T+1}$ can be obtained by $\{\mathbf{o}_t\}_{t=1}^{T+1} = D(\mathbf{V}, E(\mathbf{Q}))$, where $E(\cdot)$ and $D(\cdot)$ are transformer encoder and decoder, respectively. Using the last output $\mathbf{o}_{T+1}$ from the transformer decoder, we estimate $M$ Gaussian centers and widths by a fully connected layer followed by a Sigmoid function. Then, using the $m$-th center $\mu_m$ and the $m$-th width $\sigma_m$, we obtain the $m$-th negative proposal $\mathbf{p}_{neg}^{(m)} = [f_m(0), f_m(1), \ldots, f_m(T-1)] \in \mathbb{R}^T$ using a Gaussian function $f_m(\cdot)$:

$$f_m(t) = \exp\left(-\frac{(t/(T-1) - \mu_m)^2}{\sigma_m^2}\right). \tag{1}$$

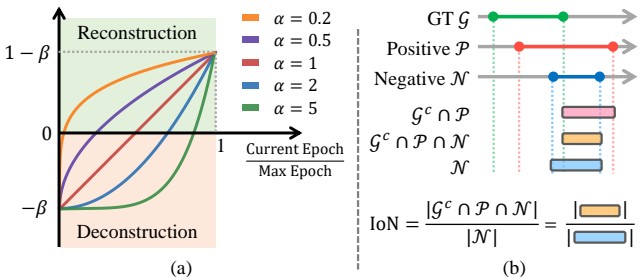

(a)

(b)

Figure 3: (a) A cross-entropy weight. As the training epoch increases, a training strategy for negative proposals is changed from the deconstruction process to the reconstruction one. (b) Intersection of Negative duration (IoN). This evaluation metric measures how well the negative proposal captures the positive proposal outside of the ground-truth boundary.

Finally, we can generate $M$ negative proposals $\{\mathbf{p}_{neg}^{(m)}\}_{m=1}^{M}$.

**Selection network.** To select useful proposals for query prediction among multiple negative proposals, we propose the selection network that determines selection weights for negative proposals. First, we use Mask-Conditioned Transformer (MCT) [28, 58] to obtain proposal-conditioned multi-modal features from a video feature $\mathbf{V}$, a masked query feature $\hat{\mathbf{Q}}$, and proposals $\{\mathbf{p}_{neg}^{(m)}\}_{m=1}^{M}$. We append a learnable token to $\hat{\mathbf{Q}}$, which is a [CLASS] token in [10]. Given $\mathbf{V}$, $\hat{\mathbf{Q}}$, and $\mathbf{p}_{neg}^{(m)}$, MCT outputs $\{\mathbf{r}_{l}^{(m)}\}_{l=1}^{L+1}$ can be obtained by $\{\mathbf{r}_{l}^{(m)}\}_{l=1}^{L+1} = D'(\hat{\mathbf{Q}}, E'(\mathbf{V}, \mathbf{p}_{neg}^{(m)}), \mathbf{p}_{neg}^{(m)})$, where $E'(\cdot)$ and $D'(\cdot)$ are MCT encoder and decoder, respectively. To focus on video segments within the area of the negative proposal, MCT uses $\mathbf{p}_{neg}^{(m)}$ as a mask for masking attention weights in every attention module of $E'(\cdot)$ and $D'(\cdot)$. More details of MCT are in [28, 58]. Using the $M$ last outputs $\{\mathbf{r}_{L+1}^{(m)}\}_{m=1}^{M}$ from $M$ negative proposals, we can estimate $M$ selection weights by two fully connected layers followed by a Softmax function. The selection weights can be written as $[s_1, s_2, \ldots, s_M]$, where $s_m \in [0, 1]$ for all $m$. Then, we multiply the $m$-th negative proposal $\mathbf{p}_{neg}^{(m)}$ by the $m$-th selection weight $s_m$, where weighted negative proposals are given by $\{s_m \mathbf{p}_{neg}^{(m)}\}_{m=1}^{M}$.

### 3.3 Dual-signed Cross-entropy Loss

**Query prediction.** Following reconstruction-based methods [28, 36], we predict a new sentence query and calculate a cross-entropy loss between the predicted query and the original query. First, using the same process of the Mask-Conditioned Transformer (MCT) in the selection network, we can obtain the MCT outputs. The difference is that we use the weighted negative proposals $s_m \mathbf{p}_{neg}^{(m)}$ as input instead of $\mathbf{p}_{neg}^{(m)}$. We feed the MCT outputs to a fully connected layer followed by a Softmax function and attain the probability scores for words in a predicted query. Then, for the $m$-th weighted negative proposal, we can calculate the cross-entropy loss $\mathcal{L}_{ce}(s_m \mathbf{p}_{neg}^{(m)})$ between the original query and the predicted query from the $m$-th weighted negative proposal.

**Deconstruction and reconstruction.** To train the negative proposals to be gradually changed from easy to hard ones, we propose

a dual-signed cross-entropy loss, which is controlled by a cross-entropy weight $c$. First, we multiply the cross-entropy losses by $c$ which is scheduled to increment from a minus value to a plus one as the training epoch progresses. According to $c$, our negative proposals are trained through two processes: the deconstruction process ($c < 0$) and the reconstruction process ($c > 0$). The deconstruction process maximizes the cross-entropy losses, which causes the negative proposals to yield predicted queries that are not relevant to an original query. As a result, the negative proposals during the deconstruction process capture a query-irrelevant temporal boundary and become easy negative proposals. In contrast, the reconstruction process minimizes the cross-entropy losses, which causes the negative proposals to yield predicted queries that are similar to an original query. As a result, the negative proposals during the reconstruction process capture a somewhat query-relevant temporal boundary and become hard negative proposals. The cross-entropy weight $c$ is scheduled by

$$c = \left(\frac{e}{e_{max}}\right)^{\alpha} - \beta, \qquad (2)$$

where $e$ is the current epoch value, $e_{max}$ is the max epoch value, and $\alpha$ and $\beta$ are a speed factor and a threshold factor, respectively, which are hyperparameters for the cross-entropy weight. As shown in Fig. 3 (a), the speed factor $\alpha$ controls the speed of changing from the deconstruction process to the reconstruction process. We can make various designs (i.e., constant, logarithmic, linear, and exponential) of the cross-entropy weight by varying the speed factor $\alpha$. The threshold factor $\beta$ acts as a threshold between the deconstruction process and the reconstruction process. Finally, we calculate the dual-signed cross-entropy loss as

$$\mathcal{L}_{dce} = c \sum_{m=1}^{M} \mathcal{L}_{ce}(s_m \mathbf{p}_{neg}^{(m)}). \qquad (3)$$

### 3.4 Training and Inference

**Training.** The overall network is trained with three losses: 1) the cross-entropy loss for a positive proposal, 2) the dual-signed cross-entropy loss $\mathcal{L}_{dce}$ for negative proposals, and 3) the multi-triplet loss $\mathcal{L}_{mt}$. The total loss can be written as $\mathcal{L} = \mathcal{L}_{ce}(\mathbf{p}_{pos}) + \lambda_1 \mathcal{L}_{dce} + \lambda_2 \mathcal{L}_{mt}$, where $\lambda_1$ and $\lambda_2$ are hyperparameters to control the balance of losses. We minimize the cross-entropy loss of the positive proposal to make the positive proposal reconstruct the sentence query. To discriminate the positive proposal from multiple negative proposals, we use the triplet loss [43] and define a multi-triplet loss $\mathcal{L}_{mt}$ that is composed of multiple triplet losses, which can be written as

$$\mathcal{L}_{mt} = \sum_{m=1}^{M} \max(\mathcal{L}_{ce}(\mathbf{p}_{pos}) - \mathcal{L}_{ce}(s_m \mathbf{p}_{neg}^{(m)}) + \gamma, 0), \qquad (4)$$

where $\gamma$ is a hyperparameter for a margin. The purpose of the multi-triplet loss $\mathcal{L}_{mt}$ is to train only the positive proposal to be discriminated from multiple negative proposals, and thus we freeze the network for negative proposal generation while minimizing the multi-triplet loss.

**Inference.** During the inference, since only a positive proposal is required to predict a query-relevant temporal boundary for the video moment localization task, our negative proposals are not used.

We follow the inference strategies of either CNM [58], CPL [59], or PPS [20] depending on the used existing network for positive proposal generation. To produce a temporal boundary from a Gaussian proposal with the center $\mu$ and width $\sigma$, starting time and ending time of the boundary are set to $\mu - \sigma/2$ and $\mu + \sigma/2$, respectively.

## 4 EXPERIMENT

### 4.1 Datasets

**Charades-STA dataset** [13] has 16,128 pairs of a video and a sentence query, which split into 12,408 training data and 3,720 testing data.

**ActivityNet Captions dataset** [23] has 71,953 pairs of a video and a sentence query, which split into 37,417 training data, 17,505 validating data ($val_1$), and 17,031 validating data ($val_2$). Following previous methods [55], we use $val_2$ as a testing set.

**TV show Retrieval dataset** [25] has 109K pairs of a video and a sentence query, which split into 87.2K training data, 10.9K validating data, and 10.9K testing data. Following previous methods [51], we use the validating data for evaluation.

### 4.2 Evaluation Metrics

We use two conventional evaluation metrics introduced in [13], which are R@$n$,IoU=$m$ and R@$n$,mIoU. The R@$n$,IoU=$m$ measures the percentage of having at least one of the top-$n$ predicted temporal boundaries with temporal Intersection over Union (tIoU) larger than the threshold $m$. The R@$n$,mIoU measures the mean value of the highest tIoU in the $n$ predicted temporal boundaries.

These two metrics only evaluate the quality of the positive proposal. To evaluate the quality of the negative proposal, we propose a new evaluation metric, **Intersection of Negative duration (IoN)**. Since there is no ground truth for good negative proposals, our IoN measures how well a negative proposal captures a poorly-generated positive proposal that fails to find a ground truth temporal boundary. If the poorly-generated positive proposals are well captured (overlapped) by negative proposals, we can learn better positive proposals through contrastive learning between positive proposals and negative proposals. Therefore, to evaluate the quality of the negative proposal, our IoN quantifies the extent to which a negative proposal captures a poorly-generated positive proposal. Given sets of the ground truth boundary $\mathcal{G}$, positive proposal boundary $\mathcal{P}$, negative proposal boundary $\mathcal{N}$, we define the IoN as

$$\text{IoN} = \frac{|\mathcal{G}^c \cap \mathcal{P} \cap \mathcal{N}|}{|\mathcal{N}|}, \qquad (5)$$

where $\mathcal{G}^c$ is the complement of $\mathcal{G}$ and $|\cdot|$ denotes the cardinality of a set. An example of IoN is depicted in Fig. 3 (b). For evaluation on datasets, we calculate **recall rates of mean IoNs (R@$n$,mIoN)** that is the mean value of the highest IoN in $n$ predicted boundaries.

### 4.3 Implementation Details

For video segment features, we use C3D [40] in ActivityNet Captions dataset and I3D [7] in Charades-STA and TV show Retrieval datasets. The maximum number of sampled video segments is 200. We employ transformers with three layers having four heads. The maximum length of sentence queries and the feature dimension $C$ are set to 20 and 256, respectively. In the randomly masked sentence

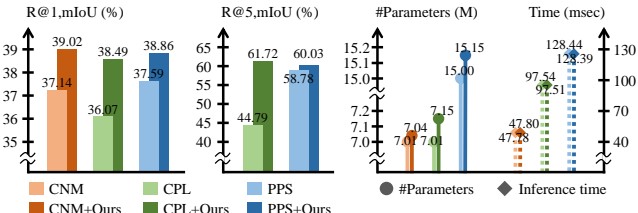

**Figure 4: Performance comparisons of the existing networks and the existing networks trained with our method on ActivityNet Captions. Our negative proposals can greatly boost the performance of the existing networks. We also measure inference time and the number of parameters. Our method only requires a negligible amount of parameters and no additional inference time.**

query, a third of the words are masked. During training, we utilize the Adam optimizer [21] with a learning rate of 0.0004. A mini-batch size is 32. Training epochs are 30 for ActivityNet Captions and 50 for Charades-STA and TV show Retrieval. We set hyperparameters as $M = 3$, $\alpha = 0.5$, $\beta = 0.8$, $\gamma = 0.15$, $\lambda_1 = 0.03$, and $\lambda_2 = 1$. More implementation details are provided in the supplementary material.

### 4.4 Comparison with State-of-the-Arts

To validate the effectiveness of our proposed method, we conduct performance comparisons between our method and previous weakly supervised moment localization methods. We use CNM [58], CPL [59], or PPS [20] for a positive generator. The performance of CNM at R@5 is not provided because CNM only generates one positive proposal while CPL and PPS generate multiple positive proposals. Fig. 4 shows that our method can boost the performance of existing methods. Especially, our negative proposals greatly improve CPL with a **16.93%** gain at R@5,mIoU. This implies that our negative proposals can contribute to the generation of high-quality positive proposals through contrastive learning. Moreover, our method increases the number of parameters by a negligible amount, as shown in Fig. 4. This is because we share the parameters of transformers for positive and negative proposals. Additional parameters are only the parameters of fully connected layers for the negative proposal generation and selection network. During the inference, only a positive proposal is exploited and thus our negative proposal is not used. Therefore, we verify that our method requires no additional inference costs, as shown in Fig. 4.

For comparisons with state-of-the-art methods, we use PPS for a positive generator. In Tabs. 1 and 2, our method surpasses most of the state-of-the-art methods on three datasets (*i.e.*, Charades-STA, ActivityNet Captions, and TV show Retrieval). For video and text encoders, while other methods use 3D ConvNet and Glove features, IRON uses an OATrans [42] and DistilBERT [34]. For a fair comparison, following IRON, we have implemented Ours[†] by replacing our encoders in Sec. 3.1 with OATrans and DistilBERT. As shown in Tab. 1, Ours[†] makes state-of-the-art performance on both Charades and ActivityNet.

**Table 1: Performance comparisons on Charades-STA and Activity Captions. Bold and underlined numbers denote the best results and the second-best results, respectively.**

| Method | Charades-STA | | | | | | ActivityNet Captions | | | | | |
| | R@1 | | | R@5 | | | R@1 | | | R@5 | | |
| | IoU=0.3 | IoU=0.5 | IoU=0.7 | IoU=0.3 | IoU=0.5 | IoU=0.7 | IoU=0.1 | IoU=0.3 | IoU=0.5 | IoU=0.1 | IoU=0.3 | IoU=0.5 |
|---|---|---|---|---|---|---|---|---|---|---|---|---|
| Random | 20.12 | 8.61 | 3.39 | 68.42 | 37.57 | 14.98 | 38.23 | 18.64 | 7.63 | 75.74 | 52.78 | 29.49 |
| CTF [9] | 39.80 | 27.30 | 12.90 | - | - | - | 74.20 | 44.30 | 23.60 | - | - | - |
| SCN [28] | 42.96 | 23.58 | 9.97 | 95.56 | 71.80 | 38.87 | 71.48 | 47.23 | 29.22 | 90.88 | 71.56 | 55.69 |
| WSTAN [44] | 43.39 | 29.35 | 12.28 | 93.04 | 76.13 | 41.53 | 79.78 | 52.45 | 30.01 | 93.15 | 79.38 | 63.42 |
| BAR [46] | 44.97 | 27.04 | 12.23 | - | - | - | - | 49.03 | 30.73 | - | - | - |
| MARN [36] | 48.55 | 31.94 | 14.81 | 90.70 | 70.00 | 37.40 | - | 47.01 | 29.95 | - | 72.02 | 57.49 |
| CCL [57] | - | 33.21 | 15.68 | - | 73.50 | 41.87 | - | 50.12 | 31.07 | - | 77.36 | 61.29 |
| RTBPN [56] | 60.04 | 32.36 | 13.24 | 97.48 | 71.85 | 41.18 | 73.73 | 49.77 | 29.63 | 93.89 | 79.89 | 60.56 |
| LoGAN [38] | 51.67 | 34.68 | 14.54 | 92.74 | 74.30 | 39.11 | - | - | - | - | - | - |
| CRM [14] | 53.66 | 34.76 | 16.37 | - | - | - | 81.61 | 55.26 | 32.19 | - | - | - |
| VCA [45] | 58.58 | 38.13 | 19.57 | 98.08 | 78.75 | 37.75 | 67.96 | 50.45 | 31.00 | 92.14 | 71.79 | 53.83 |
| LCNet [50] | 59.60 | 39.19 | 18.87 | 94.78 | 80.56 | 45.24 | 78.58 | 48.49 | 26.33 | 93.95 | 82.51 | 62.66 |
| CWSTG [8] | 43.31 | 31.02 | 16.53 | 95.54 | 77.53 | 41.91 | 71.86 | 46.62 | 29.52 | 93.75 | 80.92 | 66.61 |
| CNM [58] | 60.39 | 35.43 | 15.45 | - | - | - | 78.13 | 55.68 | 33.33 | - | - | - |
| CPL [59] | 66.40 | 49.24 | 22.39 | 96.99 | 84.71 | 52.37 | 82.55 | 55.73 | 31.37 | 87.24 | 63.05 | 43.13 |
| CPI [22] | 67.64 | 50.47 | 24.38 | 97.18 | 85.66 | 52.98 | - | - | - | - | - | - |
| CCR [29] | 68.59 | 50.79 | 23.75 | 96.85 | 84.48 | 52.44 | 80.32 | 53.21 | 30.39 | 91.44 | 71.97 | 56.50 |
| UGS [15] | 69.16 | 52.18 | 23.94 | - | - | - | 82.10 | 58.07 | **36.91** | - | - | - |
| SCANet [51] | 68.04 | 50.85 | 24.07 | 98.24 | 86.32 | 53.28 | **83.62** | 56.07 | 31.52 | 94.36 | 82.34 | 64.09 |
| OmniD [3] | 68.30 | 52.31 | 24.35 | - | - | - | 83.24 | 57.34 | 31.60 | - | - | - |
| MMDist [4] | 68.90 | **53.29** | 25.27 | - | - | - | 83.11 | 58.69 | 32.52 | - | - | - |
| PPS [20] | 69.06 | 51.49 | 26.16 | 99.18 | 86.23 | 53.01 | 81.84 | 59.29 | 31.25 | 95.28 | 85.54 | 71.32 |
| **Ours** | **70.74** | 53.04 | **26.69** | **99.24** | **90.03** | **53.86** | 83.56 | **59.71** | 33.48 | **95.50** | **86.02** | **71.63** |
| IRON† [6] | 70.71 | 51.84 | 25.01 | 98.96 | 86.80 | 54.99 | **84.42** | 58.95 | 36.27 | 96.74 | 85.60 | 68.52 |
| **Ours†** | **71.44** | **53.07** | **26.35** | **99.18** | **90.28** | **55.26** | 84.29 | **60.14** | **37.18** | **96.93** | **87.09** | **72.45** |

Unlike other methods, a method with † uses OATrans [42] and DistilBERT [34] for pre-trained encoders.

**Table 2: Performance comparisons on TV show Retrieval. Bold and underlined numbers denote the best results and the second-best results, respectively.**

| Method | R@1 | | | R@5 | | |
| | IoU=0.1 | IoU=0.3 | IoU=0.5 | IoU=0.1 | IoU=0.3 | IoU=0.5 |
|---|---|---|---|---|---|---|
| TGA [32] | 17.61 | 2.38 | 0.97 | 48.63 | 11.54 | 5.32 |
| CPL [59] | 33.16 | 7.28 | 2.11 | 64.41 | 17.93 | 8.56 |
| PPS [20] | 36.89 | 10.81 | 4.05 | 65.20 | 18.35 | 9.44 |
| SCANet [51] | 37.51 | 10.76 | 4.24 | 67.47 | 20.32 | 10.21 |
| Ours | **38.32** | **12.39** | **5.87** | **67.51** | **22.08** | **12.45** |

**Table 3: Comparisons of different cross-entropy weight designs for training negative proposals on the ActivityNet Captions.**

| Cross-entropy weight design | Speed factor $\alpha$ | R@1 | | R@5 | |
| | | IoU=0.3 | mIoU | IoU=0.3 | mIoU |
|---|---|---|---|---|---|
| Constant | 0 | 39.57 | 26.96 | 73.55 | 47.36 |
| Logarithmic | 0.2 | 53.25 | 34.70 | 81.53 | 59.39 |
| | 0.5 | **59.12** | **38.49** | **85.81** | **61.72** |
| Linear | 1 | 56.92 | 36.21 | 83.46 | 58.96 |
| Exponential | 2 | 54.31 | 33.36 | 85.35 | 57.14 |
| | 5 | 50.38 | 31.49 | 83.68 | 56.05 |

## 4.5 Ablation Study

We conduct ablation studies to analyze the impact of various components in our method. For the ablation studies, we use CPL [59] as a positive generator for computational efficiency.

**Impact of dual-signed cross-entropy loss.** As shown in Tab. 3, we conduct an experiment with different cross-entropy weight designs (*i.e.*, 'Constant', 'Logarithmic', 'Linear', and 'Exponential') by varying the speed factor $\alpha$ in Eq. (2). Here, we fix the threshold factor to 0.8. Tab. 3 verifies the effectiveness of our curriculum design. Using 'Logarithmic' ($\alpha = 0.5$) improves performance by a large margin compared to not using a curriculum design (*i.e.*, 'Constant'). The margins are **11.53%** and **14.36%** at R@1,mIoU and

R@5,mIoU, respectively. Also, 'Logarithmic' makes the best result, meaning that training negative proposals with a cross-entropy weight that changes rapidly at early training stages is most effective.

As an extension of this experiment, we conduct an experiment with different combinations of the speed factor $\alpha$ and threshold factor $\beta$ in Fig. 5. We observe the following results. First, the cross-entropy weight with ($\alpha = 0.5, \beta = 0.8$) makes the best result, which leads to the most appropriate transitions between our deconstruction and reconstruction process. Second, $\beta$ should be set higher than 0 because low $\beta$ causes the hard negative proposal to reconstruct

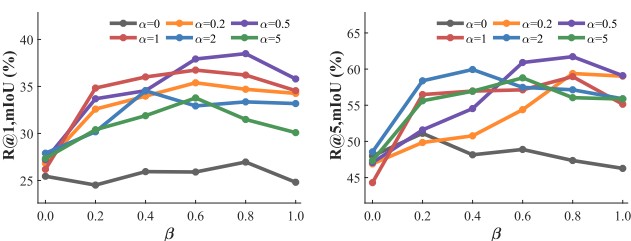

Figure 5: Results of varying the hyper-parameters for cross-entropy weights on ActivityNet Captions.

Table 4: Comparisons of different types of negative proposals on the ActivityNet Captions.

| Negative proposal | R@1 | | R@5 | |
|---|---|---|---|---|
| | IoU=0.3 | mIoU | IoU=0.3 | mIoU |
| None | 38.41 | 24.81 | 68.82 | 46.27 |
| Random | 34.46 | 23.09 | 61.56 | 40.35 |
| Rule-based square | 39.23 | 27.25 | 72.96 | 46.10 |
| Rule-based G. | 49.09 | 34.87 | 69.14 | 48.48 |
| Rule-based reversed G. [51, 58] | 50.65 | 34.54 | 69.85 | 50.19 |
| Rule-based variable-sized G. [15, 59] | 55.73 | 36.07 | 63.05 | 44.79 |
| Ours (Learning-based) | **59.12** | **38.49** | **85.81** | **61.72** |

G.: Gaussian

the query very well, which overlaps with the role of the positive proposal. The hard negative proposal should capture a somewhat query-relevant temporal boundary, not a very query-relevant temporal boundary. Third, to change the negative proposals from easy to hard ones, conducting two processes ($0 < \beta < 1$) is more effective in most cases than conducting only the reconstruction process ($\beta = 0$) or only the deconstruction process ($\beta = 1$). Fourth, using a fixed cross-entropy weight (*i.e.*, $\alpha = 0$) is not as good as the varying cross-entropy weight (*i.e.*, $\alpha \neq 0$). This result verifies the effectiveness of our dual-signed cross-entropy loss.

**Comparisons with other negative proposals.** The negative proposals used for comparisons are as follows: 'Random': a proposal having a value of zero at the location of a randomly chosen area and a value of one otherwise, 'Rule-based square': a proposal having a value of zero at the location of a positive proposal and a value of one otherwise, 'Rule-based Gaussian': two proposals of Gaussians whose location is predefined outside both sides of a positive Gaussian proposal, 'Rule-based reversed Gaussian': a proposal of Gaussian that is reversed upside down by subtracting a positive Gaussian proposal from a value of one, which is proposed in [58], and 'Rule-based variable-sized Gaussian': 'Rule-based Gaussian' whose size and location are controlled slightly by the current training epoch, which is proposed in CPL [59]. More details of these rule-based negative proposals are explained in the supplementary material. Tab. 4 shows that our learning-based negative proposals perform much better than the rule-based ones. We observe that rule-based ones only improve the performance marginally at R@5 from using no negative proposal ('None'). Unlike the rule-based ones, our negative proposals can significantly increase the performance at R@5 as well as R@1.

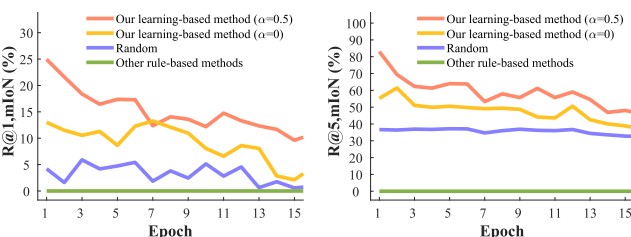

Figure 6: Performance comparisons of different negative proposals on the proposed R@n,mIoN on ActivityNet Captions.

To validate the quality of negative proposals, we use our newly proposed evaluation metric, R@$n$,mIoN, which measures how well a negative proposal captures a poorly-generated positive proposal, which is defined in Sec. 4.2. As shown in Fig. 6, we measure the performance at R@$n$,mIoN of our learning-based negative proposals and rule-based ones over the training epochs. The result shows that our learning-based negative proposals can capture a poorly-generated positive proposal while the rule-based negative proposals capture none. This is because the rule-based ones including [15, 51, 58, 59] are always defined to exist outside of positive proposals. Therefore, these rule-based ones have limitations in capturing various confusing locations because confusing locations also exist inside poorly-generated positive proposals. By capturing various confusing locations, our learning-based negative proposals have higher quality than the rule-based ones, which leads to significant performance improvement, as shown in Tab. 4.

Especially in the early training stage when the network is less trained, many poorly-generated positive proposals are generated thus it is important for negative proposals to capture the poorly-generated positive proposals. Fig. 6 shows high mIoN of our method at the early training stage which means our learning-based negative proposals can capture many poorly-generated positive proposals at the early training stage. Also, our learning-based negative proposals using a varying cross-entropy weight ($\alpha = 0.5$) make a better performance at both R@1,mIoN and R@5,mIoN than using a fixed cross-entropy weight ($\alpha = 0$). This result verifies our dual-signed cross-entropy loss with curriculum design can generate a higher quality of negative proposals. Using a fixed cross-entropy weight ($\alpha = 0$) performs better than randomly generated negative proposals ('Random'), showing that our learning-based negative proposals without the dual-signed cross-entropy loss still capture the poorly-generated positive proposals effectively.

**Freezing negative proposal for contrastive learning.** We analyze the effect of freezing negative proposals for contrastive learning in Eq. (4). As shown in Tab. 5a, it is more effective to freeze the negative proposals and only train the positive proposal through contrastive learning. Freezing the negative proposals can effectively discriminate the positive proposal from multiple negative proposals because the network can focus on training the positive proposal while the negative proposals are fixed.

**Ablations on selection strategies.** In the selection network, we use different selection strategies to select useful proposals for query prediction among multiple negative proposals. Our selection strategies are as follows: 'None': select none (no negative proposal is

**Table 5: Ablation studies on ActivityNet Captions. (a) The effect of freezing negative proposals for contrastive learning. (b) Comparisons of different selection strategies. (c) Comparisons of different numbers of negative proposals.**

| Contrastive Learning Strategy | R@1,mIoU | R@5,mIoU |
|---|---|---|
| Not freezing. negative proposal | 37.32 | 60.25 |
| Freezing negative proposal | **38.49** | **61.72** |

(a)

| Selection Strategy | R@1, mIoU | R@5, mIoU |
|---|---|---|
| None | 25.45 | 43.71 |
| Random | 32.62 | 55.54 |
| Uniform | 35.73 | 60.28 |
| Hard | 34.81 | 55.63 |
| Soft | **38.49** | **61.72** |

(b)

| #negative | R@1,mIoU | R@5,mIoU |
|---|---|---|
| 0 (None) | 25.45 | 43.71 |
| 1 | 33.84 | 56.53 |
| 2 | 35.29 | 58.71 |
| 3 | **38.49** | **61.72** |
| 4 | 38.17 | 61.63 |
| 5 | 37.88 | 61.45 |

(c)

Query: *The steamer is shown again steaming the wood floor and then the woman again, and she shows different aspects of the steamer being demonstrated.*

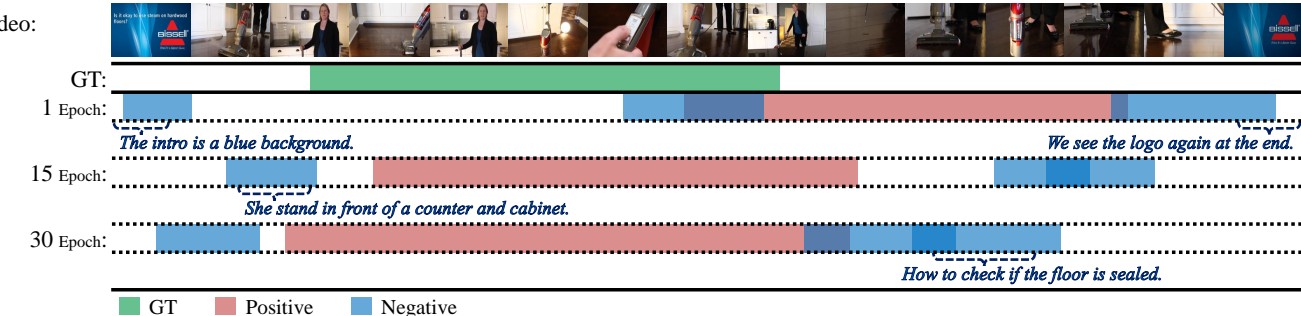

**Figure 7: Qualitative results of our negative proposals changing from easy to hard ones. We visualize the ground truth temporal boundary (Green), positive proposals (Red), and negative proposals (Blue) as the training epoch progresses. The blue texts describe the events that are not relative to the given sentence query, which can be regarded as events for the negative proposals.**

used), 'Random': randomly select one, 'Uniform': select all with the same selection weights, 'Hard': select one with the highest learnable selection weight, and 'Soft': select all with different learnable selection weights. Tab. 5b shows the following results. First, considering every proposal with different selection weights ('Soft') is useful for query prediction, making a higher performance than other strategies. Second, our negative proposal chosen at random ('Random') is still more effective than using no negative proposal ('None'). Third, considering all proposals ('Uniform') rather than just one ('Hard') is useful for query prediction.

**Number of negative proposals.** Tab. 5c shows that three negative proposals are enough to capture various confusing locations. We observe that too many negative proposals can overlap each other and become redundant.

### 4.6 Qualitative Results

We visualize our negative proposals as the training epoch progresses in Fig. 7. Here, we visualize the ground truth temporal boundary, the predicted temporal boundary from positive proposals and negative proposals as the training epoch progresses. At the early training stage, our negative proposals can capture events for easy negative, such as "The intro is a blue background" and "We see the logo again at the end". As the training epoch progresses, our negative proposals can capture events for harder negative, such as "She stands in front

of a counter and cabinet" and "How to check if the floor is sealed". By capturing confusing locations described by the various events, our negative proposals can achieve higher performance than the previous negative proposals in Tab. 4. More qualitative results are provided in the supplementary material.

## 5 CONCLUSION

In this paper, we propose learning-based negative proposals which are trained using a novel dual-signed cross-entropy loss to capture various confusing locations for weakly supervised video moment localization. Unlike the previous rule-based negative proposals, our negative proposals are 1) learnable, 2) softly selected, and 3) gradually changed from easy to hard ones through our dual-signed cross-entropy loss. Leveraging multiple contrastive losses, a positive proposal is discriminated from multiple negative proposals. In addition, we measure how well a negative proposal captures a poorly-generated positive proposal with the newly proposed evaluation metric, called Intersection of Negative duration, which proves the better quality of our learning-based negative proposals than the previous rule-based negative proposals. We also demonstrate that our negative proposals can be applied with negligible additional parameters and no inference costs, achieving state-of-the-art performance.

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
