# OpenReview forum: "Learnable Negative Proposals Using Dual-Signed Cross-Entropy Loss for Weakly Supervised Video Moment Localization"
_acmmm.org/ACMMM/2024/Conference — MM2024 Poster_

### Official Review · Reviewer_x1PK · 2024-05-12

**Rating:** 5
**Confidence:** 2

**Summary:**

This paper focuses on the Weakly Supervised Video Moment Localization task. Different from existing methods, this paper proposes learning-based negative proposals with a dual-signed cross-entropy loss. Furthermore, a novel weight schedule is introduced for the dual-signed cross-entropy loss to learn harder negative proposals gradually. Additionally, a new IoN metric is proposed to evaluate the quality of the generated negative proposals.

**Strengths:**

1. This paper proposes learnable negative proposals to enhance the training process, which is novel and effective. \
2. The illustrations are precise and appropriate, facilitating readers to quickly grasp the proposed method. \
3. The experiments are sufficient and rigorous, which demonstrate the effectiveness of each design in the proposed method. \
4. Fig.7 visually demonstrates the principle behind the effectiveness of the method, which involves gradually learning more confusing negatives to enhance the accuracy of localization.

**Limitations:**

1. Further explanation is needed for setting the selection weight for each proposal and the training process of the selection network.

**Suitability:**

3

---

### Official Review · Reviewer_gjRm · 2024-05-19

**Rating:** 2
**Confidence:** 4

**Summary:**

This paper discusses the task of video moment localization, which aims to locate the precise temporal boundaries of video segments based on natural language sentence queries. A novel dual-signed cross-entropy loss method is proposed, which improves localization accuracy by training negative proposals from easy to hard. Experimental results show that this method outperforms existing weakly supervised methods on multiple datasets

**Strengths:**

1. It proposes dual-signed cross-entropy loss method that improves the accuracy of video moment localization by training negative proposals from easy to hard.
2. Experimental results show that this method outperforms existing weakly supervised methods on multiple datasets, significantly enhancing performance.

**Limitations:**

1. The motivation of the article is unclear. As mentioned in the introduction, the negative proposals proposed in [51, 58] have limitations in capturing various confusing locations. Since the negative proposals in these methods are also learnable, please explain what specific limitations these methods have.

2. How is the positive proposal generated? Considering that both the negative proposal and the positive proposal in the reconstruction process minimize the cross-entropy loss between the predicted query and the original query, how do you distinguish between the positive and hard negative queries?

3. For experiments using I3D features on the Anet dataset, the metrics R1@0.1 and R1@0.5 significantly lag behind existing methods.

**Suitability:**

3

---

### Official Review · Reviewer_XvQT · 2024-05-25

**Rating:** 4
**Confidence:** 4

**Summary:**

This paper proposes a learning-based negative proposal generation method using a dual-signed cross-entropy loss for the task of weakly-supervised video moment localization. This loss function guides the negative proposals to focus on query-irrelevant segments in the early stages of training and on more relevant and confusing segments in the later stages. The proposed method introduces negligible additional parameters and inference costs while improving performance when integrated with previous methods. The authors also propose a new evaluation metric that measures the capability of negative proposals to capture poorly generated positive proposals.

**Strengths:**

1. The paper proposes a curriculum loss guided negative proposal generation method, verified through comprehensive comparison and ablation studies.
2. The paper is well-written and easy to follow. Figure 2 helps the understanding of the proposed method.

**Limitations:**

1. More comparisons are required; only Figure 6 includes mIoN. As mIoN is claimed as a core contribution, additional comparisons should be made using the proposed mIoN metric. Besides, Figure 6 includes only rule-based methods, which score 0 under this metric. Though it is carefully designed, it is unclear whether the metric is effective without including more methods that incorporate overlapping positive and negative proposals.
2. In the late stages of training, as the positive proposals become more precise, it is unclear how the harder negative proposals overlap with the positive ones. Figure 6 only illustrates the early stage of training. Additionally, it is important to explore any correlation between mIoN and overall performance to better understand the impact of the proposed metric.

**Suitability:**

3

---

### Official Review · Reviewer_sqpm · 2024-05-26

**Rating:** 5
**Confidence:** 3

**Summary:**

This paper introduces a novel weakly supervised video segment localization method, which is trained by introducing learnable negative proposals, gradually from easy to difficult selection, and dual-signal cross-entropy loss to capture various confusing locations in the video. Furthermore, the paper also introduces a new evaluation metric, which verifies that the quality of the negative proposals is superior to previous negative proposals, and demonstrates that the negative proposals significantly improve the performance of existing methods, while the additional parameters and inference cost are almost negligible, and achieves state-of-the-art results on different datasets.

**Strengths:**

1. Instead of promoting positive proposals, generating better negative proposals provides a new view of this problem under contrastive learning based pipeline. It is also reasonable to start with the simple negatives and gradually progress to the difficult. This idea is interesting and valuable.
2. The design of the Dual-signed Cross-entropy Loss is delicate and reasonable. Besides, the ablation study verfies the effectiveness of it.
3. The newly introduced IoN metric further improves the study from the view of negative proposal learning. This metric can be used in future works.
3. The presented figure 7 clearly demonstrates how the learning processes with the help of designed negative learning.

**Limitations:**

1. The performance of the method is not consistantly good across different settings and datasets. The increase margin compared with other method can be mere or even be minus sometimes.
2. A concern is that how to prevent the negative generated from becoming too confusing with the GT positive, when the learning is undergoing the reconstruction learning. Simply relying on the hyper parameters can limit the method's ability to generalize on other datasets.

**Suitability:**

3

---

### Meta-Review · Area_Chair_k25u · 2024-06-30

**Recommendation:** Accept (Poster)
**Confidence:** 5

**Metareview:**

The AC goes through the paper, rebuttal and review comments. This paper got 4 acceptances. After discussion, all the reviewers acknowledge the good performance and the novelty of the proposed loss, metric as well as negative proposal. Thus, the AC recommends accepting this paper.